# Reliability of Self-Reported Height and Weight in Children: A School-Based Cross-Sectional Study and a Review

**DOI:** 10.3390/nu15010075

**Published:** 2022-12-23

**Authors:** Magali Rios-Leyvraz, Natalia Ortega, Arnaud Chiolero

**Affiliations:** 1Population Health Laboratory (#PopHealthLab), University of Fribourg, 1700 Fribourg, Switzerland; 2Bern Institute of Primary Health Care (BIHAM), Faculty of Medicine, University of Bern, 3012 Bern, Switzerland; 3Department of Epidemiology, School of Population and Global Health, McGill University, Montréal, QC H3A 1G1, Canada

**Keywords:** weight, height, body mass index, children, review, Switzerland

## Abstract

Since anthropometric measurements are not always feasible in large surveys, self-reported values are an alternative. Our objective was to assess the reliability of self-reported weight and height values compared to measured values in children with (1) a cross-sectional study in Switzerland and (2) a comprehensive review with a meta-analysis. We conducted a secondary analysis of data from a school-based study in Switzerland of 2616 children and a review of 63 published studies including 122,629 children. In the cross-sectional study, self-reported and measured values were highly correlated (weight: *r* = 0.96; height: *r* = 0.92; body mass index (BMI) *r* = 0.88), although self-reported values tended to underestimate measured values (weight: −1.4 kg; height: −0.9 cm; BMI: −0.4 kg/m^2^). Prevalence of underweight was overestimated and prevalence of overweight was underestimated using self-reported values. In the meta-analysis, high correlations were found between self-reported and measured values (weight: *r* = 0.94; height: *r* = 0.87; BMI: *r* = 0.88). Weight (−1.4 kg) and BMI (−0.7 kg/m^2^) were underestimated, and height was slightly overestimated (+0.1 cm) with self-reported values. Self-reported values tended to be more reliable in children above 11 years old. Self-reported weight and height in children can be a reliable alternative to measurements, but should be used with caution to estimate over- or underweight prevalence.

## 1. Introduction

Weight and height are key indicators of growth and health in children, especially in a context of rising obesity [1]. In large surveys, the measurement of weight and height is often not feasible due to financial, logistic, and human resources limitations and self-reported values are used as an alternative. Easy to collect, self-reported values are prone to misreporting [2,3]. In adults, weight tends to be underestimated and height tends to be overestimated [4,5]. As a result, body mass index (BMI) is underestimated which leads to underestimation in the prevalence of overweight and obesity [5] and to biased estimates of the risk of obesity-related health outcomes [6,7].

To evaluate how reliable self-reported weight and height values are, it is necessary to compare self-reported and measured values in the population of interest because evidence suggest that reporting bias can differ between populations depending on cultural norms [8,9,10]. Estimating the degree of reporting bias is necessary to use self-reported instead of measured values. In Switzerland, bias in the self-report of weight and height has been well documented in adults [5], but not in children and adolescents.

Our objective was therefore to assess the reliability of self-reported weight and height in a large school-based sample of children in Switzerland and to compare our findings with other studies that have assessed the reliability of self-reported weight and height in children.

## 2. Materials and Methods

### 2.1. School-Based Cross-Sectional Study

#### 2.1.1. Data Collection

We conducted a secondary analysis of a school-based cross-sectional study, that took place in the canton of Vaud, in Switzerland between September 2005 and May 2006 [11,12]. All children from sixth grade in public schools in Vaud, Switzerland were invited to participate in the study. Ethical approval was obtained from the Ethical Committee of the Canton of Vaud (approval number 91/05). Consent was sought from the directors of the schools. Signed informed consent of the child and one parent were obtained.

The children were asked their height and weight, and were then measured with fixed stadiometers (at 0.1 cm) and with precision electronic scales (at 0.1 kg) [13]. Children were in light clothes and without shoes and measurement were made by trained clinical assistants. Basic information of the participants’ socio-economic and lifestyle characteristics (i.e., sex, physical activity, TV viewing, parental education, and parental BMI) were collected by questionnaire. The questionnaires are provided in Appendix A of the Appendix A.

#### 2.1.2. Data Analysis

The body mass index (BMI) was calculated as the weight in kilogram (kg) divided by the squared height in meter (m). BMI z-scores were calculated based on the age- and gender-specific reference values from the 2000 Centers for Disease Control and Prevention (CDC) [14] and were classified into underweight (BMI z-score <−2), normal weight (BMI z-score −2 to 1) and overweight/obese (BMI z-score >1) categories. The Shapiro–Wilk test confirmed that the hypothesis that the data of the variables under study came from the normal distribution. Mean differences were calculated by subtracting measured values to self-reported values. Pearson’s *r* correlations were computed to estimate the correlation between self-reported and measured values. Intra-class correlations were calculated to estimate agreement between self-reported and measured values. Kappa statistics were computed to estimate the agreement between BMI categories (underweight, normal weight, and overweight). Distributions were plotted in histograms. The agreement between measures were plotted with Bland–Altman plots [15]. Differences between groups were assessed with t tests. The statistical analyses were conducted with R (version 3.6.3, R Foundation for Statistical Computing, Vienna, Austria) in the graphical user interface RAnalyticFlow (version 3.1.8, Ef-prime, Inc., Tokyo, Japan).

### 2.2. Review

#### 2.2.1. Data Collection

We conducted a comprehensive non-systematic review to identify other studies having assessed the reliability of self-reported weight and height in children for comparison with the results of our school-based study. Studies which assessed the reliability of self-reported weight and height in children by parents were not included. We searched for published studies comparing self-reported and measured weight and height in children in Medline using the keywords “child”, “height”, “weight”, “self-reported”, and “measured”. We completed our search by screening the reference lists of the studies included and reviews on related subjects [15,16,17,18]. Cross-sectional and cohort studies comparing self-reported and measured weight, height and BMI in apparently healthy children up to 18 years of age (mean age) were included.

#### 2.2.2. Data Analysis

The following information were extracted: country, sample size, sex, age, measurement method, time between self-report and measure, missing data, Pearson’s *r* correlations, intra-class correlations, mean differences (self-reported—measured), and percentage differences between reported and measured weight, height and BMI. Data transformations and imputations were done according to the Cochrane Handbook for Systematic Reviews of Interventions [19]. When possible, data were merged and pooled reliability was estimated using random effects meta-analyses. Sub-group analyses were conducted to compare results by sex, age (6–11, 12–15, 16–21 years old), region (Asia, Australasia, Europe, Latin America, North America), knowing about subsequent measurement (whether the participants knew that they were going to be measured and weighed after reporting their weight and height), time between reported and measured values (< or ≥7 days), and self-reported data collection method (paper questionnaire, on-line questionnaire, in-person interview, telephone interview). The statistical analyses were conducted with R (version 3.6.3, R Foundation for Statistical Computing, Vienna, Austria) in the graphical user interface RAnalyticFlow (version 3.1.8, Ef-prime, Inc., Japan).

## 3. Results

### 3.1. Cross-Sectional Study

#### 3.1.1. Participant Characteristics

Out of the 6873 children invited in the study, 5334 (78%) agreed to participate. Out of the 5334 children who participated, 2620 (49%) reported values for weight and height and 5212 (98%) had their weight and height measured. A total of 2616 (49%) children had both reported and measured weight and height values and were included in the analysis. The children who did not report weight or height did not differ to those who did in age and height, but had a higher weight (2.9 kg, 95% CI 2.2, 3.5) and a higher BMI (0.06 kg/m^2^, 95% CI 0.4, 0.8). The characteristics of the children included in the analysis are shown in Table 1.

#### 3.1.2. Differences between Self-Reported and Measured Values

The distributions of the self-reported and measured values are shown in Figure 1 and the differences between these values are presented in Table 2. Mean self-reported weight and height were lower than measured values, by −1.4 kg (95% CI −1.5, −1.3) and −0.9 cm (95% CI −1.0, −0.8), respectively. This translated into a slightly lower mean BMI (−0.36 kg/m^2^; CI 95% −0.41, −0.31) and mean BMI z-scores (−0.14; 95% CI −0.16, −0.12) based on self-reported data. The Pearson correlations were 0.96 for weight, 0.92 for height, 0.88 for BMI, and 0.84 for BMI z-score. The Bland–Altman plots are shown in Figure 2. The limits of agreement (LOA) were −3.8 and 6.6 kg for weight, −5.4 and 7.1 cm for height, −2.2 and 2.9 kg/m^2^ for BMI, and −0.93 and 1.21 for BMI z-score.

Based on measured values, 85.2% of the children had a normal weight, 2.5% were underweight, and 12.2% were overweight/obese. Based on self-reported values, 85.9% of the children had a normal weight, 3.8% were underweight and 10.3% were overweight/obese. With self-reported measures, the prevalence of underweight was overestimated (+1.3%) and the prevalence of overweight was underestimated (−1.9%). The percentage of children misclassified to an incorrect BMI category as a result of inaccurate self-reported height or weight was 9.4% (*n* = 247). Among the children which were underweight (*n* = 66), 41% were incorrectly classified as normal weight (*n* = 27). Among the children which were overweight (*n* = 320), 33% were incorrectly classified as normal weight (*n* = 105). Among children with normal weight (*n* = 2230), 5% were incorrectly classified as under- or overweight (*n* = 115). The Cohen’s kappa for two BMI categories (non-overweight and overweight) was 0.70 and for three BMI categories (underweight, normal weight, and overweight) 0.63, which are both considered moderate agreement.

#### 3.1.3. Digit Preferences

The distributions in Figure 1 indicated peaks of self-reported values around the 0 and 5 last digits. The distribution of the last digits is shown in Figure 3. For the measured values, every digit was reported approximately 10% of the time. The last digits 0 and 5 of the self-reported values were reported much more often, i.e., for weight 21% and 15%, respectively, and for height 27% and 16%, respectively. Rounding was done more often towards a lower value weight than towards a higher value.

#### 3.1.4. Factors Associated with Reporting

Some participant characteristics were associated with differences in reporting (Table 3). Differences in reporting were found according to BMI category: overweight/obese children underestimated more their weight in comparison with children with normal weight, and children with underweight tended to overestimate their weight in a larger magnitude. No significant differences were found for height according to BMI category.

### 3.2. Review

#### 3.2.1. Study Characteristics

A total of 63 studies (including the current study) comparing self-reported and measured weight and height in children were identified (see Table 4 and Appendix A). 

The studies included 122,629 participants who were between 6 and 21 years of age with a mean of 14 years. The studies were conducted in Europe (*n* = 26), North America (*n* = 23), Asia (*n* = 8), Australasia/Oceania (*n* = 4), and South America (*n* = 2).

A total of 37 studies reported percentage of non-response which ranged from 0 to 80%, with an average of approximately 25%. Among the 51 studies which reported the time between the self-reported and measured values were taken, 32 (63%) measured the values on the same day, 10 (20%) between 2 and 7 days, and the remaining (17%) between the day after and up to 6 months after the self-reported values. Among the 27 studies which reported whether the participants knew if they were going to be measured subsequently, in 12 (44%) studies the participants knew and in 15 (56%) they did not know. The self-reported values were collected by paper questionnaires in 46 (73%) studies, online questionnaires in 4 (6%) studies, in-person interview in 11 (17%) studies and phone interview in 2 (3%) studies.

The Pearson’s correlations for weight, height and BMI, respectively were reported in 33, 33 and 31 studies and ranged from 0.72 to 0.99, from 0.32 to 0.99, and from 0.49 to 0.98, respectively. The mean differences between self-reported and measured values for weight, height and BMI, respectively were reported in 45, 48, and 34 studies and ranged from −7 to 0.6 kg, −10 to 8 cm, −6.5 to 2.0 kg/m^2^. The differences in the prevalence of overweight, obesity, and overweight/obesity, respectively were reported in 20, 22, and 19 studies and ranged from −15% to 3%, −11% to 2%, and −11% to 13% (differences in percentage points).

Three studies included participants from specific ethnicities (i.e., Melanesian [49] and American Indians [53,54]) and showed markedly lower correlations and higher mean differences than the other included studies. A study including solely overweight children [61] also found markedly lower correlations and greater underestimation of weight. One study compared child and parental reports [52] and found that parental reports are slightly more accurate that child reports. The children reported their weight and height while at home in three studies [24,30,35]. Fourteen studies included questions on body weight perception, past weight loss efforts, and confidence in self-reported values [3,25,32,37,39,42,46,49,52,58,60,63,68,73]. Thirteen studies developed a conversion factor or equation to correct the self-reported values [32,37,42,44,49,50,51,55,57,60,63,79].

#### 3.2.2. Meta-Analysis

The results of the meta-analysis are shown in Table 5. Self-reported weight was highly correlated with measured weight (*r* = 0.94), with a systematic underestimation (−1.4 kg; 95% CI −1.5, −1.2). Self-reported height showed a slightly lower, although still high, correlation with measured values (*r* = 0.87). In some studies, height was underestimated, while in others it was overestimated, which resulted on average in a small mean difference (0.1 cm; 95% CI 0.0, 0.1). Based on self-reported values, BMI were highly correlated with measured BMI (*r* = 0.87) and were on average underestimated (−0.7 kg/m^2^; 95%CI −1.0, −0.3).

There were no substantial differences between boys and girls in how weight was reported, but there were for height, for which boys tended to underestimate height more than girls (−0.3 cm, 95% CI −0.5, −0.2, vs. 0.0 cm, 95% CI −0.1, 0.1). There were differences across age categories. The correlations between self-reported and measured height and BMI increased with age (6–11 years: *r* = 0.84, 95% CI 0.79, 0.88, and *r* = 0.86, 95% CI 0.82, 0.90; 12–15 years: *r* = 0.87, 95% CI 0.85, 0.90, and *r* = 0.87, 95% CI 0.86, 0.89; 16–21 years: *r* = 0.92, 95% CI 0.90, 0.94, and *r* = 0.92, 95% CI 0.90, 0.95). The mean difference between self-reported and measured height increased with age, going from under- to over-estimation (6–11 years: −0.5 cm, 12–15 years: −0.2, 16–21 years: 1.4 cm). The differences between self-reported and measured weights between age groups were not significantly different. The differences between regions were significant, suggesting that in some regions, Europe especially, the self-reported values are more reliable than in other regions. 

Using the metric (kg and cm) or the imperial (lb and inch) system was not associated with the degree of bias.

When the participants knew that they were going to be weighed and measured after their self-report, they tended to report more strongly correlated values for both weight (*r* = 0.96 vs. 0.92) and height (*r* = 0.93 vs. *r* = 0.87) and closer values for weight (−1.1 kg vs. −1.9 kg), but not for height. Having self-reported and measured values done close together in time (less than one week vs. one week or more), resulted in smaller mean differences for weight (−1.3 vs. −2.3 kg). There were no differences in correlations between data collection methods, but there were differences in the mean differences for weight, height and BMI where the smallest differences were with in-person interviews (−0.7 kg, −0.3 cm and −0.3 kg/m^2^).

## 4. Discussion

In this large sample of children in Switzerland, self-reported weight and height were highly correlated with measured values. Self-reported weight and height tended to be lower than measured values and resulted in lower BMI. Misclassification of children in both extremes of BMI occurred, due to overestimation of weight in underweight children and underestimation in overweight children. There was a tendency to round to 0 and 5 digits. The review showed similar findings, with high correlations, and with underestimated weight and BMI, but not height. Self-reported weight and height in children can be a reliable alternative to in-person measurements but should be used with caution to estimate the prevalence of over- or underweight.

The age of the respondents plays an important role in the reliability of self-reported weight and height. Several studies reported that children 7–11 years old provided less reliable reported values than older children [21,27,29]. In our review, the correlation between the reported and measured BMI was the lowest in the age group 6–11 years. In this age group, the weight was more underestimated than in the older age groups. Height showed an interesting pattern: height under-estimated in the younger 6–11-year-olds, the most accurate in the 12–15-year-olds, and over-estimated in the older 16–21-year-olds. In the 6–11-year-olds, the children could be less capable of reporting height or likely to have undergone a growth spurt since the last time measured. In the 16–21-year-olds, the overestimation of height matches what has been found typically in adults [4,5].

The reliability of self-reported values differed slightly across regions. Hence, the reliability of self-reported values tended to be higher in European regions, and this is consistent with the results of our study in Switzerland. One might also hypothesize that the differences between regions found in the meta-analysis could be explained by differences in body image in different cultures and by the inclusion of some studies in certain regions (i.e., North America and Australasia/Oceania) with especially low reliability due to participants’ characteristics. In fact, in some low-resources populations or in studies including a large share of children of low socio-economic background [49,53,54], participants could have been weighed and measured less often, and therefore less aware of their weight and height. Moreover, in populations with a high prevalence of overweight/obesity and/or underweight, self-reported values could be on average less reliable, because overweight and underweight children are more likely to misreport their weight, as also shown in the current Swiss study and review.

The design of the study also affected the reliability of reporting. In fact, children who knew that they were going to be measured afterwards, reported more accurate values. They perceived that misreporting would be discovered. Children who underwent in-person interviews also reported more accurate values. They probably perceived that the person interviewing could guess by seeing them whether misreporting occurred. In the present Swiss study, the children answered paper questionnaires and knew that they were going to be measured afterwards.

In the present study, conducted in Switzerland, rounding towards the end digits zero and five was found likewise in adults [8]. One study suggested that the effect of rounding could be bigger using the imperial rather than the metric system [58]. However, differences in reporting bias between imperial and the metric system were not found in our review. We hypothesized that rounding towards certain digits could differ between cultures (e.g., 4 and 7), but we have not found any literature on the subject.

One major issue with self-reported values in children is the high percentage of non-reporting, with the proportion of non-reporting largely different from one study to the other. We can hypothesize two reasons for non-reporting: not knowing the values or not answering to avoid stigma, especially in underweight and overweight children [3,21,34,37]. Younger children less frequently know their weight and height compared to older children [21,34]. In our study, children with missing self-reported values tended to be more overweight than those with complete values. Some researchers have tried to address this issue by imputing missing values and were able to provide more accurate estimates of the prevalence of overweight [2].

To overcome the previous limitations, several studies have proposed correction equations to improve the reliability of self-reported values [32,37,44,49,50,51,55,57,60,79]. Some included a variable on the body perception (e.g., too thin, just right, too fat) [32,49,60]. A study included two questions (on perceived ability to report weight/height and weighing/measuring history) to identify children with low response capability [3]. As children grow rapidly and undergo growth spurts, if the last time the child was measured or weighed occurred some time ago, the self-reported values could be less reliable. Another study suggested that, if questionnaires are completed at home, children could be advised to weighed themselves and be measured before answering if they are unsure [58,77]. A study suggested that adding the parent’s report on their child’s weight and height could be helpful [52]. Weight loss history is another variable which could be useful [58]. We found one study that corrected for missing values [2], but none that corrected for rounding.

Our school-based cross-sectional study has several strengths. It included many children through a school-based recruitment which covers nearly all children in the region (96% of the children attend school), and self-reported and measured values were collected on the same day. A limitation of this study is that it collected neither information on body image perception, which could have been useful to develop a correction factor for the self-reported values nor on how reliable the participants thought their self-reported values were. Another limitation of this study is that the samples consisted of children from one region in Switzerland, a country with high resources. The results of this study might therefore not be applicable to less affluent regions of the world or in younger children. However, thanks to the accompanying extensive review, we were able to put the results in context of other studies.

The strengths of this review were the inclusion of a number of studies worldwide and the multiple sub-group meta-analysis, allowing the identification of the population and study characteristics that ensure the best reliability of self-reported values. We did not identify any other reviews that also compared the correlations and mean differences between self-reported and measured anthropometrics in children and adolescents worldwide. One review compared these measures among children in studies conducted in the United States [18]. Two other reviews compared measures among children in studies conducted worldwide, but only to estimate how reliable was the prevalence and diagnosis of overweight and obesity [16,17]. Although our review was not systematic, it was comprehensive and included more studies than the previous reviews on related subjects [15,16,17,18].

## 5. Conclusions

In conclusion, self-reported weight and height in children can be a cheap and reliable alternative to in-person measurements. The reliability of self-reported values can be increased using correction equations that include for example time since last weighed/measured, confidence in reported values, perceived body image, parental report, weight loss history. This can be especially useful in specific groups of children (younger, underweight and overweight children) that are known to provide less reliable or more missing self-reported values and for the estimation of the prevalence of over- and underweight.

## Figures and Tables

**Figure 1 nutrients-15-00075-f001:**
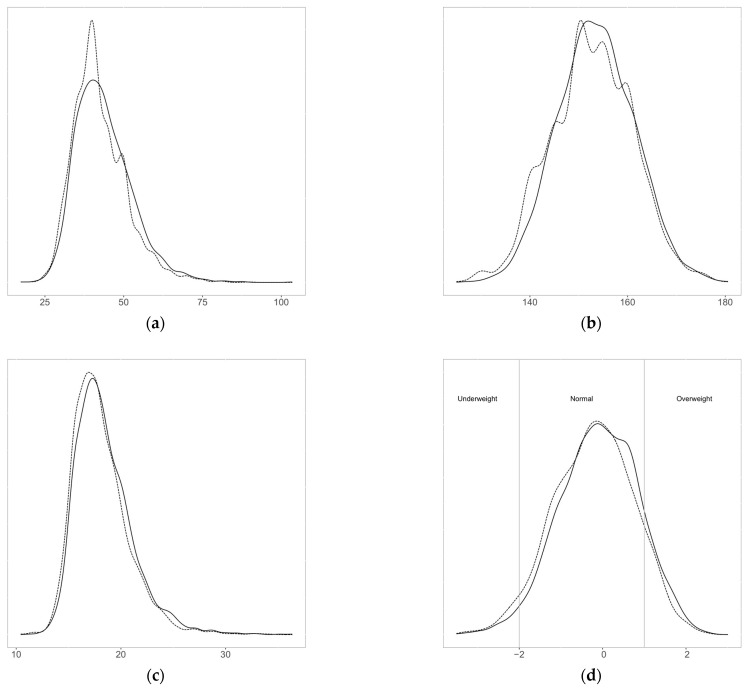
Distribution of (**a**) weight (kg), (**b**) height (cm), (**c**) BMI (kg/m^2^), and (**d**) BMI z-score (continuous line: measured, dashed line: self-reported).

**Figure 2 nutrients-15-00075-f002:**
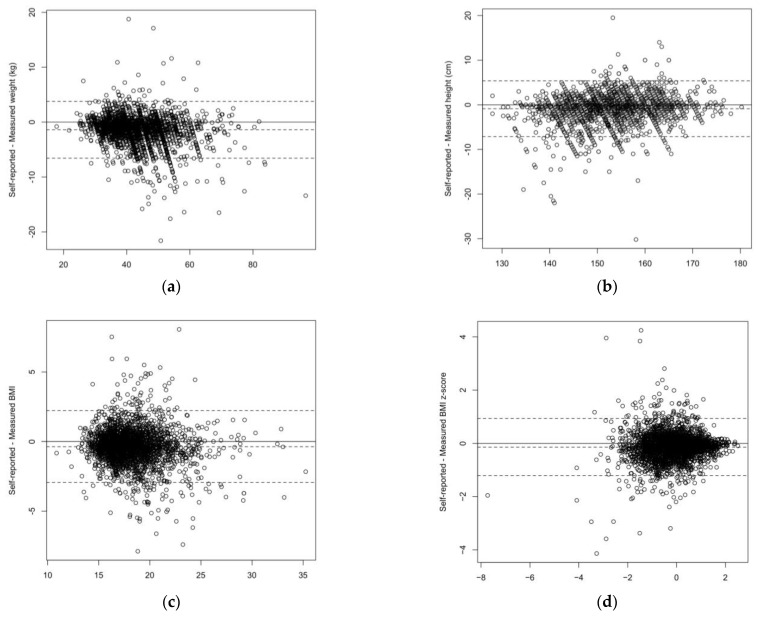
Bland–Altman plots for (**a**) weight (kg), (**b**) height (cm), (**c**) BMI (kg/m^2^), and (**d**) BMI z-score (dashed lines: Mean difference and 95% confidence interval; each circle represent one individual).

**Figure 3 nutrients-15-00075-f003:**
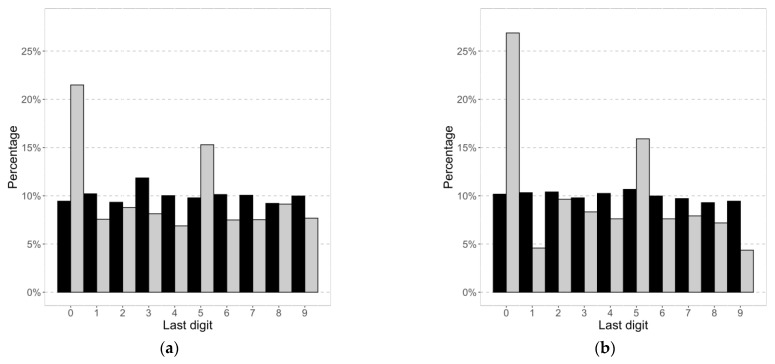
Distribution of last digit for (**a**) weight (kg) and (**b**) height (cm) for self-reported (white) and measured (black) values.

**Table 1 nutrients-15-00075-t001:** Participant characteristics (*n* = 2616).

Variable	Group	Mean ± SD (Range) or Percentage (*n*)
Age (years)	All	12.2 ± 0.5 (10.2–14.8)
Sex	Female	47.5% (1243)
Physical activity	Never	4.2% (110)
	A few times per month	8.0% (210)
	1–2× per week	37.8% (962)
	3–4× per week	27.6% (722)
	5–6× per week	10.0% (261)
	Daily	13.1% (344)
	Missing	0.3% (7)
TV viewing during week	Never	15% (390)
	<15 min per day	15% (401)
	15–30 min per day	25% (652)
	30–60 min per day	26% (670)
	1–2 h per day	14% (360)
	>2 h per day	5% (119)
	Missing	1% (24)
Nationality	Swiss	79% (1846)
	Other	22% (568)
	Missing	8% (202)
Highest parental education	University	32% (828)
	High school baccalaureate	14% (368)
	Apprenticeship	34% (895)
	Primary/secondary school	10% (263)
	Other/missing	10% (260)
Parental BMI (kg/m^2^)	Mother	22.9 ± 3.6 (14.5–44.98)
	Father	25.4 ± 3.2 (14.6–42.9)

**Table 2 nutrients-15-00075-t002:** Self-reported and measured weight, height, BMI and BMI z-scores, mean, mean difference, and correlations.

Measures	Self-Reported (Mean ± SD)	Measured (Mean ± SD)	Mean Difference * (Mean (95% CI))	Pearson’s *r* Correlation	Intra-Class Correlation
Weight (kg)	42.3 ± 8.4	43.7 ± 9.0	−1.4 (−1.5, −1.3)	0.96	0.94
Height (cm)	152.7 ± 8.4	153.5 ± 7.8	−0.9 (−1.0, −0.8)	0.92	0.92
BMI (kg/m^2^)	18.07 ± 2.63	18.43 ± 2.76	−0.36 (−0.41, −0.31)	0.88	0.87
BMI z-score	−0.23 ± 0.99	−0.09 ± 0.96	−0.14 (−0.16, −0.12)	0.84	0.83

* Mean difference = self-reported—measured.

**Table 3 nutrients-15-00075-t003:** Mean differences (95% CI) between self-reported and measured weight and height in difference population groups.

Variable	Group	Weight (kg)	Height (cm)
Sex	Boys	−1.3 (−1.5, −1.2)	−0.9 (−1.1, −0.7)
	Girls	−1.5 (−1.6, −1.4)	−0.9 (−1.0, −0.7)
BMI category	Underweight	0.8 (−0.1, 1.6)	−1.2 (−2.0, −0.4)
	Normal	−1.2 (−3.2, −1.1)	−0.9 (−1.1, −0.8)
	Overweight	−3.2 (−3.7, −2.9)	−0.3 (−0.7, 0.0)

**Table 4 nutrients-15-00075-t004:** Review of studies comparing self-reported and measured weight, height and BMI in children.

Study ID	Country	Sample Size	Sex	Age (Years) Mean ± SD (Range)	Participants Aware of Upcoming Measurement	Time between Self-Reported and Measured Values	No Self-Reported Data	Collection of Self-Reported Values
Aasvee 2015 [20]	Estonia	3379	Both	11, 13, 15	Yes	Same day	19%	Paper questionnaire
Abalkhail 2002 [21]	Saudi Arabia	2860	Both	14 ± 3 (9–21)	Unclear	Same day	59%	In-person interview
Abraham 2004 [22]	Australia	683	Female	15 + 2 (11–18)	Unclear	Same day	Unclear	Paper questionnaire
Ambrosi-Randic 2007 [23]	Croatia	234	Female	(10–18)	Unclear	Same day	Unclear	In-person interview
Andersen 2005 [24]	Norway	159	Both	8, 12	Unclear	3 d	3%	Paper questionnaire
Bae 2010 [25]	South Korea	379	Both	(8–16)	Yes	Unclear	Unclear	Online questionnaire
Baile 2014 [26]	Spain	171	Both	(10–11)	Unclear	Same day	Unclear	Paper questionnaire
Beck 2012 [27]	United States	487	Both	(6–7, 8–9, 10–11)	Unclear	Unclear	Unclear	In-person interview
Beghin 2013 [28]	Europe	3865	Both	(12.5–17.5)	Unclear	Same day	5–23%	Paper questionnaire
Berg 2001 [29]	Sweden	628	Both	9, 12, 15, 18	Unclear	Same day	Unclear	Paper questionnaire
Brault 2015 [30]	Canada	875	Both	(8–12)	Unclear	Unclear	Unclear	Paper questionnaire
Brener 2003 [31]	United States	2032	Both	(14–18)	Unclear	0–2 d	23%	Paper questionnaire
Brettschneider 2015 [32]	Germany	3468	Both	(11–17)	Yes	Same day	80%	In-person interview
Brooks-Gunn 1987 [33]	United States	151	Female	(11–13)	Unclear	Same day	Unclear	Paper questionnaire
Buttenheim 2013 [34]	United States	613	Both	(12–17)	Unclear	Unclear	6–23%	Online questionnaire
Chan 2013 [35]	China	1614	Both	(6–18)	Yes	1–2 weeks	21%	Paper questionnaire
Charalampos 2009 [36]	Cyprus	579	Both	15 ± 2	Unclear	Same day	Unclear	Paper questionnaire
Chau 2013 [37]	France	1559	Both	13 ± 1 (9–18)	Yes	Same day	4–8%	Paper questionnaire
Clarke 2014 [38]	United States	19,238	Both	13	Yes	Same day	Unclear	Paper questionnaire
Dalton 2014 [39]	United States	1243	Both	15 ± 1 (14–18)	Unclear	Unclear	18%	In-person interview
Davis 1994 [40]	United States	829	Both	(12–19)	Unclear	A few weeks	Unclear	Phone interview
De Vriendt 2009 [41]	Belgium	982	Both	14 ±1 (10–18)	No	Same day	1%	Paper questionnaire
Domingues 2011 [42]	Portugal	719	Both	14 (10–19)	No	Same day	1%	Paper questionnaire
Drake 2013 [2]	United States	407	Both	(12–18)	No	Unclear	75%	Phone interview
Ekström 2015 [43]	Sweden	1698	Both	16.5 ± 0.3	No	4 weeks	56%	Online questionnaire
Elgar 2005 [44]	Canada	395	Both	16 (15–17)	No	Same day	6%	Paper questionnaire
Enes 2009 [45]	Brazil	360	Both	(10–14)	No	Same day	14%	In-person interview
Farre Rovira 2002 [46]	Spain	568	Both	(14–20)	Yes	Same day	Unclear	Paper questionnaire
Fonseca 2010 [47]	Portugal	462	Both	14 ± 2 (12–16)	Unclear	Same day	0%	Paper questionnaire
Fortenberry 1992 [48]	United States	725	Both	17 (14–20)	Unclear	Same day	4%	Paper questionnaire
Frayon 2017 [49]	New Caledonia	665	Both	14 ± 1 (10–17)	Unclear	maximum 2 d	3–6%	Online questionnaire
Ghosh-Dastidar 2016 [50]	United States	475	Both	13 ± 1	Unclear	1 month	74%	Online questionnaire
Giacchi 1998 [51]	Italy	133	Both	(15–17)	Unclear	1 week	7%	Paper questionnaire
Goodman 2000 [52]	United States	11,495	Both	(12–18)	Unclear	Unclear	47%	In-person interview
Hauck 1995 [53]	United States	619	Both	15 ± 2 (12–19)	Unclear	1–7 d	28%	Paper questionnaire
Himes 1992 [54]	United States	69	Both	15 (12–19)	No	Unclear	Unclear	Paper questionnaire
Himes 2001 [55]	United States	1635	Both	(12–16)	Yes	Same day	4–16%	Paper questionnaire
Himes 2005 [56]	United States	3797	Both	15 ± 2 (12–18)	Unclear	Same or next day	4–5%	Paper questionnaire
Jansen 2006 [57]	Netherlands	499	Both	(12–14)	No	3 months	30%	Paper questionnaire
Jayawardene 2014 [58]	United States	7160	Both	(14–17)	Unclear	Unclear	31%	Paper questionnaire
Kee 2017 [59]	Malaysia	663	Both	(13–17)	No	Max 6 months	5%	Paper questionnaire
Kurth 2010 [60]	Germany	3436	Both	(11–17)	Unclear	Same day	Unclear	In-person interview
Lee 2006 [61]	United States	71	Both	13 ± 3 (8–18)	Unclear	Same day	Unclear	Paper questionnaire
Lee 2013 [62]	South Korea	422	Both	11 ± 1	Unclear	Unclear	Unclear	Paper questionnaire
Legleye 2014 [63]	France	303	Both	(17–18)	No	Same day	11%	Paper questionnaire
Linhart 2010 [64]	Israel	517	Both	(13–14)	Unclear	Unclear	44%	Paper questionnaire
Morrissey 2006 [65]	United States	416	Both	(10–16)	Yes	Same day	26%	Paper questionnaire
Ohlmer 2012 [66]	Germany	162	Female	14 ± 1 (12–16)	Unclear	Same day	Unclear	Paper questionnaire
Perez 2015 [67]	United States	24,221	Both	13.7, 16.7	Unclear	Unclear	Unclear	Paper questionnaire
Rasmussen 2007 [68]	Sweden	2726	Both	15	Unclear	Up to 1 month	11%	Paper questionnaire
Rasmussen 2013 [3]	Denmark	2100	Both	11, 13, 15	Unclear	1–3 weeks	11%	Paper questionnaire
Robinson 2014 [69]	United States	92	Both	10 ± 1	Unclear	Next day	Unclear	Paper questionnaire
Rodrigues 2013 [70]	Brazil	97	Both	16 ± 1 (14–19)	Unclear	Same day	Unclear	In-person interview
Seghers 2010 [71]	Belgium	789	Both	9 ± 1 (8–11)	Unclear	Same day	22%	Paper questionnaire
Stefan 2019 [72]	Croatia	286	Both	16 ± 1	Yes	Same day	Unclear	Paper questionnaire
Strauss 1999 [73]	United States	1932	Both	(12–16)	Yes	Same day	14%	In-person interview
Tienboon 1992 [74]	Australia	204	Both	(14–15)	No	Same day	Unclear	Paper questionnaire
Tokmakidis 2007 [75]	Greece	676	Both	11, 12	No	Next day	Unclear	Paper questionnaire
Tsigilis 2006 [76]	Greece	300	Both	16 ± 1	No	Next day	Unclear	Paper questionnaire
Wang 2002 [77]	Australia	572	Both	(15–19)	No	A few weeks	Unclear	Paper questionnaire
Yoshitake 2012 [78]	Japan	358	Both	10–11, 13–14	Unclear	Same day	Unclear	Paper questionnaire
Zhou 2010 [79]	China	1761	Both	(12–16)	No	1 week	2%	Paper questionnaire
Current study	Switzerland	2616	Both	12 ± 1 (10–14)	Yes	Same day	38%	Paper questionnaire

**Table 5 nutrients-15-00075-t005:** Results of the meta-analyses (mean (95% CI)). *p*-values are for differences between sub-groups.

	Pearson’s Correlation						Mean Difference					
Sub-Groups	Weight (kg)	*p*	Height (cm)	*p*	BMI (kg/m^2^)	*p*	Weight (kg)	*p*	Height (cm)	*p*	BMI (kg/m^2^)	*p*
All	0.94 (0.94, 0.95)		0.87 (0.86, 0.89)		0.88 (0.87, 0.89)		−1.4 (−1.5, −1.2)		0.1 (0.0, 0.1)		−0.7 (−1.0, −0.3)	
Sex												
Boys	0.95 (0.94, 0.96)	0.908	0.91 (0.89, 0.92)	0.031	0.89 (0.87, 0.90)	0.394	−1.2 (−1.5, −1.0)	0.113	−0.4 (−0.5, −0.2)	<0.001	−0.5 (−0.6, −0.4)	0.038
Girls	0.95 (0.94, 0.96)		0.88 (0.86, 0.90)		0.90 (0.88, 0.91)		−1.5 (−1.7, −1.3)		0.0 (−0.1, 0.1)		−0.6 (−0.7, −0.5)	
Age												
6–11 years	0.94 (0.92, 0.96)	0.909	0.84 (0.79, 0.88)	<0.001	0.86 (0.82, 0.90)	0.002	−1.6 (−2.0, −1.1)	0.461	−0.5 (−1, 0.1)	<0.001	−0.5 (−0.8, −0.3)	0.033
12–15 years	0.93 (0.92, 0.95)		0.87 (0.85, 0.90)		0.87 (0.86, 0.89)		−1.6 (−1.8, −1.4)		−0.2 (−0.3, −0.1)		−0.8 (−1.2, −0.4)	
16–21 years	0.93 (0.89, 0.97)		0.92 (0.90, 0.94)		0.92 (0.90, 0.95)		−1.4 (−1.7, −1.1)		1.4 (0.9, 2.0)		−0.9 (−1.1, −0.7)	
Region												
Asia	0.94 (0.90, 0.98)	<0.001	0.89 (0.83, 0.96)	<0.001	0.89 (0.82, 0.97)	<0.001	−1.2 (−1.8, −0.5)	0.230	−0.3 (−0.8, 0.2)	0.125	0.0 (−1.1, 1.2)	<0.001
Australasia/ Oceania	0.87 (0.84, 0.90)		0.77 (0.71, 0.83)		0.67 (0.59, 0.75)		−1.8 (−2.7, −0.9)		−0.6 (−2.3, 1.2)		−0.3 (−0.5, 0.0)	
Europe	0.95 (0.94, 0.96)		0.93 (0.92, 0.94)		0.89 (0.88, 0.91)		−1.4 (−1.7, −1.2)		0.2 (0.1, 0.3)		−0.9 (−1.1, −0.7)	
North America	0.94 (0.93, 0.95)		0.82 (0.78, 0.85)		0.87 (0.85, 0.89)		−1.3 (−1.5, −1.0)		0.0 (−0.6, 0.6)		−0.9 (−1.5, −0.3)	
South America							−0.8 (−1.4, −0.2)		−1.2 (−3.6, 1.2)		0.0 (−0.2, 0.2)	
Knowing about subsequent measurement												
Yes	0.96 (0.95, 0.97)	0.016	0.93 (0.91, 0.95)	<0.001	0.91 (0.88, 0.93)	0.022	−1.1 (−1.3, −0.8)	0.007	−0.3 (−0.6, 0.0)	0.018	−0.8 (−1.9, 0.2)	0.894
No	0.92 (0.90, 0.94)		0.87 (0.84, 0.91)		0.84 (0.80, 0.88)		−1.9 (−2.4, −1.4)		0.1 (0.0, 0.2)		−0.5 (−1.3, 0.2)	
Unclear	0.94 (0.94, 0.95)		0.85 (0.82, 0.88)		0.89 (0.87, 0.9)		−1.2 (−1.4, −1.0)		0.2 (0.1, 0.3)		−0.7 (−0.9, −0.5)	
Time between self-reported and measured values												
<7 days	0.95 (0.94, 0.96)	0.165	0.87 (0.85, 0.89)	0.123	0.89 (0.87, 0.90)	0.472	−1.3 (−1.5, −1.1)	0.001	0.1 (0.0, 0.2)	0.010	−0.7 (−0.9, −0.6)	0.766
≥7 days	0.92 (0.90, 0.95)		0.91 (0.88, 0.93)		0.84 (0.76, 0.91)		−2.3 (−3.1, −1.6)		−0.3 (−0.5, −0.1)		−0.3 (−1.5, 0.9)	
Unclear	0.95 (0.93, 0.96)		0.86 (0.79, 0.92)		0.89 (0.86, 0.91)		−0.8 (−1.1, −0.6)		0.2 (−0.3, 0.6)		−0.8 (−1.3, −0.2)	
Self-report data collection												
Paper questionnaire	0.94 (0.94, 0.95)	0.606	0.88 (0.85, 0.90)	0.454	0.88 (0.87, 0.89)	0.148	−1.4 (−1.6, −1.2)	0.005	0.2 (0.1, 0.2)	<0.001	−0.8 (−1.3, −0.2)	0.001
Online questionnaire	-		-		-		−1.9 (−2.7, −1.1)		−0.7 (−1.5, 0.1)		−0.5 (−0.6, −0.4)	
In-person interview	0.94 (0.92, 0.95)		0.86 (0.82, 0.90)		0.89 (0.86, 0.92)		−0.8 (−1.2, −0.4)		−0.4 (−0.7, 0.2)		−0.3 (−0.4, −0.2)	
Phone interview	0.94 (0.93, 0.95)		0.86 (0.84, 0.88)		0.86 (0.84, 0.88)							

## Data Availability

The data of the review are provided in Appendix A of the the Appendix A.

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
