# Peer review of "Reliability of Self-Reported Height and Weight in Children: A School-Based Cross-Sectional Study and a Review"

_nutrients, 2022, doi:10.3390/nu15010075_

Round 1

Reviewer 1 Report

Rios-Leyvraz, et. al, conducted a study in which they compared the Self-Reported weight, height, and BMI, in children, with the measured values, either with a secondary analysis of a study, and as a meta-analysis of literature data.

Based on their results, they concluded that self-reported data is a valid alternative to in-person measurments. These results may help to define the study design and establish best way to collect data base don the required accuracy.

Some corrections to the text are needed as reported:

Line 55-56, please remove were repeated twice

Line 246, But repeated twice

Line 282 an should be any?

Line 305 same day repated

Author Response

Reviewer 1

Rios-Leyvraz, et. al, conducted a study in which they compared the Self-Reported weight, height, and BMI, in children, with the measured values, either with a secondary analysis of a study, and as a meta-analysis of literature data.

Based on their results, they concluded that self-reported data is a valid alternative to in-person measurments. These results may help to define the study design and establish best way to collect data based on the required accuracy.

We thank the reviewer for the thorough review and for catching the typo errors below.

Some corrections to the text are needed as reported:

Line 55-56, please remove were repeated twice

We have removed the repeated word.

Line 246, But repeated twice

We have removed the repeated word.

Line 282 an should be any?

We have corrected “an” to “any”.

Line 305 same day repated

We have removed the repeated word.

Reviewer 2 Report

The study seems interesting and may be used in population with higher levels of education.

The main concerns are:

1. The study with a sample size of 5334 is although a good sample, may not be representative of other part of the world where different measuring units and levels of education are there.

2. The reproducibility may be limited in children with no/low education or who do not remember their vitals.

3. Children tend to grow rapidly which may have little corroboration with the last measurements.

4. The authors are silent on the parameters other than anthropometry, such as laboratory values, family history, etc

Author Response

Reviewer 2

The study seems interesting and may be used in population with higher levels of education.

We thank the reviewer for the thorough review and for useful comments.

The main concerns are:

  1. The study with a sample size of 5334 is although a good sample, may not be representative of other part of the world where different measuring units and levels of education are there.

Our cross-sectional study did not aim to be representative of other parts of the world. Further, it is possible that the results could differ if the study had been conducted in another country and this is why we complemented this study with an extensive review of other studies in the world, allowing to put our results in perspective and ease their use in another context.

Measuring units were not associated with the degree of reliability of self-reported measurements, as reported in the result section on lines 225-227: “Using the metric (kg and cm) or the imperial (lb and inch) system was not associated with the degree of bias.”

We mention the difference in reliability across region of the study in the discussion section, on lines 369-380: “The reliability of the self-reported values differed slightly across regions. Hence, the reliability of self-reported values tended to be higher in European regions, and that is consistent with the results of our study in Switzerland. One might also hypothesize that the differences between regions could be explained by differences in body image in different cultures and by the inclusion of some studies in certain regions with especially low reliability due to participants’ characteristics. In fact, in some low-resources populations or in studies including a large share of children of low socio-economic background [19–21], participants could have been weighed and measured less often, and therefore less aware of their weight and height. Moreover, in populations with a high prevalence of overweight/obesity and/or underweight, self-reported values could be on average less reliable because overweight and underweight children are more likely to misreport their weight.”

To make sure that we have addressed the reviewers comment, we have also added a limitation in the discussion, on lines 443-447: “Another limitation of this study is that the samples consisted of children from one region in Switzerland, a country with high resources. The results of this study might therefore not be applicable to less affluent regions of the world or in younger children. However, thanks to the accompanying extensive review, we were able to put the results in context of other studies.”

  1. The reproducibility may be limited in children with no/low education or who do not remember their vitals.

We agree with the reviewers point and have put more emphasis in the discussion on this point.

We have added a sentence in the discussion, on lines 375-378: “In fact, in some low-resources populations or in studies including a large share of children of low socio-economic background [19–21], participants could have been weighed and measured less often, and therefore less aware of their weight and height.” And in lines 443-446: “Another limitation of this study is that the samples consisted of children from one region in Switzerland, a country with high resources. The results of this study might therefore not be applicable to less affluent regions of the world or in younger children. However, thanks to the accompanying extensive review, we were able to put the results in context of other studies.”

  1. Children tend to grow rapidly which may have little corroboration with the last measurements.

We agree with this point and have added a sentence in the discussion, on lines 428-430: “As children grow rapidly and undergo growth spurts, if the last time the child was measured or weighed occurred some time ago, the self-reported values could be less reliable.”

  1. The authors are silent on the parameters other than anthropometry, such as laboratory values, family history, etc.

We agree with the reviewer that other parameters could have been interesting to investigate. However, this was not our research question and this data was not available. 

Reviewer 3 Report

Thank you for the opportunity to revise the manuscript entitled: " Reliability of self-reported height and weight in children: 2 A school-based cross-sectional study and review".

 The manuscript is very interesting and it could be in its current form.

Please, add the Ethical Committee approval number.

The quality of the tables is very satisfactory. The reference list exhaustively covers the relevant literature in an unbiased way. Statistical methodologies are valid and coherent with the aim of the study. All the methodology section is sufficiently documented in order to allow replication studies.

The manuscript follows a high rigor in its structure, high quality in the writing and in the quality of the content.

I believe that the manuscript will capture the interest of the audience interesting in this field.

Author Response

Reviewer 3

Thank you for the opportunity to revise the manuscript entitled: " Reliability of self-reported height and weight in children: 2 A school-based cross-sectional study and review".

 The manuscript is very interesting and it could be in its current form.

Please, add the Ethical Committee approval number.

The study has been approved by the Ethic Committee of the Canton de Vaud (approval number 91/05). We have added this information on lines 54-55.

The quality of the tables is very satisfactory. The reference list exhaustively covers the relevant literature in an unbiased way. Statistical methodologies are valid and coherent with the aim of the study. All the methodology section is sufficiently documented in order to allow replication studies.

The manuscript follows a high rigor in its structure, high quality in the writing and in the quality of the content.

I believe that the manuscript will capture the interest of the audience interesting in this field.

We thank the reviewer for the thorough review and the comments.

Reviewer 4 Report

The article shows the reliability of children and adolescents to self-report their body weight and height. This has been studied previously, but the manuscript compares its results with previous studies. The writing and discourse are well developed, but need to expand the introduction and clarify various aspects of the analysis that could lead to severe errors.

Major revision:

Introduction

1. The introduction is too short and needs to be expanded. An important point to work on is the state of the art of the study: Why is this study important and relevant? What information does it add to the science?

Methods

2. Why is a BMI z-score used instead of the BMI percentile provided by the Centers for Disease Control and Prevention (CDC)? The latter is based on gender and age, whereas the BMI z-score is not, which could lead to misinterpretation.

3. What is the distribution of variables? Parametric analyses have been used in the manuscript but it is not known whether the mean is really a measure of centrality or not.

Furthermore, looking at figure 3, based on the frequency of responses, it appears that the mean is not close to the median. Therefore, it is necessary to obtain normality and, if the distribution of the variables does not meet this assumption, the analyses should be changed from parametric to non-parametric.

4. Why was Medline used and not a more restrictive database in terms of study quality, such as WOS or SCOPUS?

5. Which questionnaires were used to obtain information on physical activity, TV viewing time during the week and parental education? This information is necessary and relevant.

Author Response

Reviewer 4

The article shows the reliability of children and adolescents to self-report their body weight and height. This has been studied previously, but the manuscript compares its results with previous studies. The writing and discourse are well developed, but need to expand the introduction and clarify various aspects of the analysis that could lead to severe errors.

We thank the reviewer for the thorough review and useful comments.

Major revision:

Introduction

  1. The introduction is too short and needs to be expanded. An important point to work on is the state of the art of the study: Why is this study important and relevant? What information does it add to the science?

We argue why this study is important and relevant in the Introduction section on lines 27-34: “Weight and height are key indicators of growth and health in children, especially in a context of rising obesity [1]. In large surveys, the measurement of weight and height is often not feasible due to financial, logistic, and human resources limitations and self-reported values are used as an alternative. Easy to collect, self-reported values are prone to misreporting [2,3]. In adults, weight tends to be underestimated and height tends to be overestimated [4,5]. As a result, body mass index (BMI) is underestimated which leads to underestimation in the prevalence of overweight and obesity [5] and to biased estimates of the risk of obesity-related health outcomes [6,7].”

In the introduction section, on lines 39-40, we mention what is the research gap that our study fills: “In Switzerland, bias in the self-report of weight and height has been well documented in adults [5], but not in children and adolescents.”

Further, in the Discussion section, on lines 328-335, we mention how our review fills the current research gap: “We did not identify any other reviews that also compared the correlations and mean differences between self-reported and measured anthropometrics in children and adolescents worldwide. One review compared these measures among children in studies conducted in the United States [17]. Two other reviews compared measures among children in studies conducted worldwide but only to estimate how reliable was the prevalence and diagnosis of overweight and obesity [15,16]. Although our review was not systematic, it was comprehensive and included more studies than the previous reviews on related subjects [14–17].”

Methods

  1. Why is a BMI z-score used instead of the BMI percentile provided by the Centers for Disease Control and Prevention (CDC)? The latter is based on gender and age, whereas the BMI z-score is not, which could lead to misinterpretation.

We have used the age- and gender-specific BMI z-score from the CDC. To make this clearer to the reader, we have added this precision in the text, on lines 61-64: “BMI z-scores were calculated based on the age- and gender-specific reference values from the 2000 Centers for Disease Control and Prevention (CDC) [13] and were classified into underweight (BMI z-score <-2), normal weight (BMI z-score -2 to 1) and overweight/obese (BMI z-score >1) categories.”

  1. What is the distribution of variables? Parametric analyses have been used in the manuscript but it is not known whether the mean is really a measure of centrality or not.

Furthermore, looking at figure 3, based on the frequency of responses, it appears that the mean is not close to the median. Therefore, it is necessary to obtain normality and, if the distribution of the variables does not meet this assumption, the analyses should be changed from parametric to non-parametric.

We have reviewed the distribution of the variables. On a visual inspection (Figure 3), the distribution of the variables does not seem perfectly normal. However, the mean and medians are close to equal for all parameters. We ran Shapiro-Wilk normality tests and confirmed that all the parameters were normally distributed. We have added this in the method, on lines 66-67: “Shapiro-Wilk test confirmed the assumption of distribution normality was true for all variables.”

  1. Why was Medline used and not a more restrictive database in terms of study quality, such as WOS or SCOPUS?

We chose to run the search in Medline to be able to identify as much studies as possible. This allowed us to conduct an extensive search in a research area with relatively little literature.

  1. Which questionnaires were used to obtain information on physical activity, TV viewing time during the week and parental education? This information is necessary and relevant.

We have added the questionnaire in the Supplementary Information.

Round 2

Reviewer 4 Report

The authors have adequately answered the questions and correctly resolved the recommendations. The reviewer's opinion is that the manuscript is ready.

Author Response

We thank the reviewer for the comments.